# Child drownings in Bangladesh: need for action

Md. Jamal Hossain ,[1] Md. Al-Mamun ,[2] Morshed Alam ,[3] Mst. Rukaia Khatun,[4,5] Md. Moklesur Rahman Sarker,[1] Md. Rabiul Islam[6]

[1]Department of Pharmacy, State University of Bangladesh, Dhaka, Bangladesh
[2]Department of Sociology, Bangabandhu Sheikh Mujibur Rahman Science and Technology University, Gopalganj, Bangladesh
[3]Institution of Education and Research, Jagannath University, Dhaka, Bangladesh
[4]Department of Public Health, Varendra University, Rajshahi, Bangladesh
[5]Seba Nursing Institute, Chapainawabganj, Bangladesh
[6]Department of Pharmacy, University of Asia Pacific, Dhaka, Bangladesh

**Correspondence to**
Mr. Md. Jamal Hossain; jamal. du.p48@gmail.com

## ABSTRACT

Drowning is one of the major public health concerns, and children are the most vulnerable victims of drowning death in Bangladesh, which has been a paramount threat to child survival. Based on available data, we intend to underline the prevalence and associated risk factors for child drowning deaths in Bangladesh. According to the Center for Injury Prevention and Research, Bangladesh, about 19 000 people of all ages drown per year across the country, where approximately 77% are children (<18 years), which means that over 40 Bangladeshi children drown per day. A recent survey reported that as of data collected from January 2020 to June 2021, 83% of drowning victims were children. Insufficient parental supervision, mother's illiteracy, lack of swimming ability, male gender, children under 5 years, geographical and environmental conditions, seasonality, and disasters significantly contribute to child drowning deaths in Bangladesh. We urge the governments and local administrations to address the current crisis by coordinating and integrating several effective efforts to prevent child drowning deaths.

Drowning is a significant yet frequently overlooked public health hazard in both low-income and middle-income and high-income countries. According to the WHO Global Report-2019, 236 000 people worldwide die every year from drowning, the world's third-leading cause of injury-related death.[1] Drowning is the eighth most significant cause of mortality and the twelfth leading source of illness burden in South-East Asia, including Bangladesh.[2] Over half of all drowning deaths occur in the Western Pacific and Southeast Asian region. As of data published by UNICEF and the National Institute of Population Research and Training, drowning-death accounted for over one-fourth (26%) and two-fifths (42%) of all deaths in children aged 1 to 4 years in Bangladesh in 2003 and 2011, respectively (figure 1A).[2] Notably, children aged 0–4 years are near three times more likely to drown than children aged 10–17 years.[3] The risk of drowning is significantly higher in rural children.[2 3]

A survey conducted by the Center for Injury Prevention and Research, Bangladesh in collaboration with the Department of Health and UNICEF reported that approximately 19 000 people of all ages drown per year in Bangladesh. Among them, 14 500 (77%) are children.[4] Another recent survey performed by Society for Media and Suitable Human Communication Techniques in support of Global Health Advocacy Incubator documented 1402 deaths from 875 drowning incidents, where 83% (n=1164) of victims were children in the last one and half years (January 2020–June 2021). More than two-thirds (~ 69%, n=962) were below 9 years (0–4 years=514 and 5–9 years=448; figure 1B).[5] The study also reported that boys were significantly more at risk of being victims of drowning when compared with girls (60.82% vs 38.65%), and the Dhaka (n=322) and Chittagong (n=267) divisions showed the highest number of deaths (figure 1C).[5]

Children in low-income and middle-income countries, including Bangladesh, are more likely to drown if they are not adequately supervised, male, there are no physical barriers between them and bodies of water, and they are not proficient swimmers.[1–3] In countries and regions where social, economic, and geographical shifts occur, the risks of drowning deaths vary widely. Besides, parents' illiteracy (no schooling) was associated with 3.7 times and 2.9 times higher risk of fatal and non-fatal drowning, respectively, than secondary or higher-level education in Bangladesh.[2] Around 80% of the drowning deaths occur due to exposure to ponds, channels, buckets, and ditches within 20 m of victim's home.[2 4] Furthermore, three-fifths (60%) drowning cases occur between 09:00 and 13:00, and children of large households (five or more children) are more at risk than children of small families (less than three children).[3] Particularly, children in Bangladesh's lower regions (particularly in the southern part) are at greater risk than those in the country's higher areas because

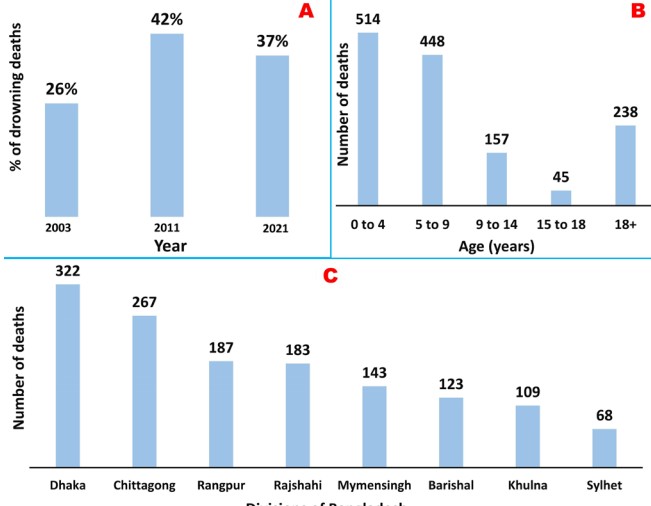

**Figure 1** (A) Percentage of drowning deaths in children aged 1–4 years (2003 and 2011) and in children under 4 years of age (2021) in Bangladesh.[2 5] (B) The distribution of drowning deaths occurred among Bangladeshi children according to age range in January 2020 to June 2021.[5] (C) The distribution of child drowning deaths among the divisions of Bangladesh in January 2020 to June 2021.[5]

of their geographic location. Moreover, lack of parental supervision and oversight, disasters, lack of awareness of water safety, and unsafe behaviour around water may be considered dominant factors behind child mortality due to drowning.[2 3]

The government of Bangladesh has already traced the issue of drowning as a prime concern of children's death and initiated some pilot actions for child protection. However, the regular epidemiological surveillance and the rigorous drive to boost awareness countrywide are still unfocused. The government and all the relevant social organisations should be committed to implementing a long-term national strategy based on proven interventions like establishing community-based childcare institutions with daycare to curb the high incidence of drowning. It is evident from a pilot-based study that community-based supervision of young children and teaching of swimming to older children reduced by 82% and 90% chances of

drowning, respectively.[6] Besides, parents and guardians need to make sure the children understand the numerous places where they could drown. Furthermore, attention and raising community awareness of proper drowning rescue and resuscitation techniques should be a vital component of any programme to reduce the death toll in Bangladesh.

**Contributors** MA-M and MJH conceived the idea, and MJH designed the study. MJH, MA-M, MA and MRK collected data. MJH, MA-M, MA and MRK drafted the original version of the manuscript. MJH, MMRS and MRI critically revised and improved the manuscript. All authors reviewed and approved the final version of the manuscript for publication.

**Funding** The authors have not declared a specific grant for this research from any funding agency in the public, commercial or not-for-profit sectors.

**Competing interests** None declared.

**Patient consent for publication** Not applicable.

**Ethics approval** Not applicable.

**Provenance and peer review** Not commissioned; externally peer reviewed.

**ORCID iDs**
Md. Jamal Hossain http://orcid.org/0000-0001-9706-207X
Md. Al-Mamun http://orcid.org/0000-0002-4133-757X
Morshed Alam http://orcid.org/0000-0001-9234-3075

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
