## [Reviewer comments · BMJ Paediatrics Open]

ARTICLE DETAILS

TITLE (PROVISIONAL)	Non-fatal child drowning death and its associated risk factors in Bangladesh: Urgent call for actionable measures
AUTHORS	Hossain, Md. Jamal Al-Mamun, Md. Alam, Morshed Khatun, Mst. Rukaia Sarker, Md. Mokleshur Rahman Islam, Md. Rabiul

VERSION 1 – REVIEW

REVIEWER	Reviewer name: Dr. Colleen Saunders Institution and Country: University of Cape Town Faculty of Health Sciences, South Africa Competing interests: None
REVIEW RETURNED	24-Mar-2022

GENERAL COMMENTS	Dear authors, Thank you for the opportunity to review this original research letter. I commend you for raising awareness of the drowning burden - particularly in a setting with such a high drowning burden. I have the following comments regarding this submission. Article format: According to the journal's guidelines for authors, Original Research Letters should present studies that present original research but address a single research question. In my opinion the current submission does not meet this description. The current submission more closely resembles an editorial or letter to the editor that synthesises evidence from several sources and presents these as a narrative review. It does not present original research. Study title: The title specifically speaks to non-fatal drowning. However the manuscript itself speaks only to the fatal drowning burden in Bangladesh and makes no specific reference to risk factors and interventions related to non-fatal drowning. The title does not, therefore, accurately reflect the contents of the manuscript. Language editing: I appreciate that you may not be writing in your first language and commend you for this. However, there are a number of instances in which the grammar and sentence structure is incorrect and changes the meaning of the statement. For example - line 80 - "riskier" versus more at risk; line 67 - "nice" versus nine. I think the manuscript would be strengthened by having a colleague who is fluent in English provide a brief language editing review. Specific comments: - Line 51-52: Please use the updated 2019 WHO estimate (236 000 people) https://www.who.int/news-room/fact-sheets/detail/drowning- Line 65-66: This statement is incorrect - "83% of children died from drowning" should actually read that 83% of drowning victims
---

	are children. - Figure 1: Please be cautious of confusing "death rate" with prevalence or "Proportion of all deaths". These are quite different measures and death rate would require a clear indication of the denominator. I wish you well as you take this manuscript forward, and look forward to reading further contributions.
--	---

REVIEWER	Reviewer name: Dr. Amy E. Peden Institution and Country: University of New South Wales Sydney, United Kingdom of Great Britain and Northern Ireland Competing interests: None
REVIEW RETURNED	15-Mar-2022

GENERAL COMMENTS	Thank you for the opportunity to review manuscript ID bmjpo--2022-001464 which is an original research letter entitled "Non-fatal child drowning death and its associated risk factors in Bangladesh: Urgent call for actionable measures" which was submitted for consideration for publication in BMJ Paediatrics Open. This research letter details the important issue of child drowning in Bangladesh and makes a call for action to reduce the risk. I offer the following feedback in the hope the letter can be further strengthened. Feedback Page 2, Line 23: I'm not sure child abuse is the most appropriate keyword for this work, I'd encourage authors to revise this. However, I also note, keywords on the first page are different to others listed elsewhere. Suggest the authors clarify and make consistent but avoid use of child abuse. Page 3 Line 29, Page 3 Line 15: The title and use of the phrase non-fatal drowning death is confusing and misleading. Suggest the authors revise – if this letter wants to convey overall drowning risk among children then suggest Child drowning and its associated risk factors.... If the specific focus is on death or non-fatal suggest using only one phrase in the title and throughout the letter, including the abstract. Page 3, Line 17 – unsure what raising concerns means in this context – suggest clarifying Page 3, Line 25 – how can a death rate be a % - suggest clarifying – do you mean between Jan 2020 and June 2021, child drowning comprises 83% of all deaths? Page 3 Line 29 and throughout paper: suggest rephrasing natural calamities to disasters Page 3, Lines 45-47: Suggest low and middle income and high income countries instead of developing and developed countries Page 3, Line 49 – not third leading cause of death, third leading cause of injury-related death Page 3 line 52 – occur not occurs Page 4 line 3 – words missing here – suggest two-fifths (42%) of a deaths... Page 4, Line 5 and throughout paper – crossreference the specific panel of the figure when you mention Figure 1 ie Figure 1, Panel A Page 4 line 21 – two-third should be two-thirds – nice should also be nine? I think you mean below 10 years if the age range is 5-9 years inclusive Page 4 line 24 – more victims doesn't make sense, suggest boys are more at risk when compared to girls Page 4 – line 39 – are happened to suggest replace with occur Page 4 line 43 – define large households – how many children is considered a large household as per the research? Page 5 – line 55 – suggest pilot actions not pilot basis actions Page 5 line 5 – focused off – not clear expression – do you mean unfocused or not given enough priority? Page 5 line 10 – a pilot based study or pilot based studies
--

VERSION 1 – AUTHOR RESPONSE

Editor in Chief Comments to Author:

Title amend to "Child drownings in Bangladesh: Need for action"

Authors responses: Dear Respected Editor-in-Chief, we would like to thank you and your assigned reviewers for reviewing the manuscript and providing your expert opinions for the improvement of the manuscript. The title of the study has been changed as follows according to your suggestions.

Child drownings in Bangladesh: Need for action

Avoid the term "Non-fatal child drowning death" throughout the paper as it is incorrect.

Authors responses: Thank you very much for the suggestion. We have removed the term "non-fatal" from the manuscript.

Use numbers not % in the text and Figures. It will make your paper easier to read

Avoid misuse of prevalence rates and death rates

Authors responses: Thank you very much for your kind suggestions. We have revised and modified the Figure as follows:

Figure 1. (A) The drowning death rate among children aged 1-4 years (2003 and 2011) and below four years (2021) in Bangladesh.^{2,5} (B) The distribution of drowning deaths occurred among Bangladeshi children according to age range in January 2020 to June 2021.⁵ (C) The distribution of child drowning deaths among the divisions of Bangladesh in January 2020 to June 2021.⁵

The English needs improving. Sorry

Authors responses: Thank you very much for the suggestions. We have carefully revised the whole manuscript and eradicated any unintentional grammar and typos errors. Besides, an English language expert has revised the manuscript for its language improvement.

Reviewer: 1

Dr. Amy E. Peden, University of New South Wales Sydney

Comments to the Author

Thank you for the opportunity to review manuscript ID bmjpo--2022-001464 which is an original research letter entitled "Non-fatal child drowning death and its associated risk factors in Bangladesh: Urgent call for actionable measures" which was submitted for consideration for publication in BMJ Paediatrics Open. This research letter details the important issue of child drowning in Bangladesh and makes a call for action to reduce the risk. I offer the following feedback in the hope the letter can be further strengthened.

Authors responses: Thank you very much for reviewing the manuscript and providing your expert opinions for the improvement of the manuscript.

Feedback

Page 2, Line 23: I'm not sure child abuse is the most appropriate keyword for this work, I'd encourage authors to revise this. However, I also note, keywords on the first page are different to others listed elsewhere. Suggest the authors clarify and make consistent but avoid use of child abuse.

Authors responses: Thank you very much for the points regarding the keywords. We did not use child abuse as keyword rather we used child drowning as a keyword.

Page 3 Line 29, Page 3 Line 15: The title and use of the phrase non-fatal drowning death is confusing and misleading. Suggest the authors revise – if this letter wants to convey overall drowning risk among children then suggest Child drowning and its associated risk factors.... If the specific focus is on death or non-fatal suggest using only one phrase in the title and throughout the letter, including the abstract.

Authors responses: Thank you very much for your suggestions. We have revised the title and changed it as follows-

Child drownings in Bangladesh: Need for action

Page 3, Line 17 – unsure what raising concerns means in this context – suggest clarifying

Authors responses: Thank you very much for your point. We have revised and eradicated the confusion as follows:

Based on available data, we intend to underline the prevalence and associated risk factors for child drowning deaths in Bangladesh. (Lines: 34-35)

Page 3, Line 25 – how can a death rate be a % - suggest clarifying – do you mean between Jan 2020 and June 2021, child drowning comprises 83% of all deaths?

Authors responses: Thank you very much for your question. We have revised the lines as follows:

A recent survey reported that as of data collected from January 2020 to June 2021, 83% of drowning victims were children. (Lines: 38-40)

Page 3 Line 29 and throughout paper: suggest rephrasing natural calamities to disasters

Authors responses: Thank you very much for your suggestion. We have rephrased the natural calamities to disasters throughout the manuscript.

Page 3, Lines 45-47: Suggest low and middle income and high income countries instead of developing and developed countries

Authors responses: Thank you very much for your suggestion. We have revised the lines as follows:

Drowning is a significant yet frequently overlooked public health hazard in both low and middle income and high-income countries. (Lines: 49-50)

Page 3, Line 49 – not third leading cause of death, third leading cause of injury-related death

Authors responses: Thank you very much for the nice suggestion. We have revised the lines as follows:

According to the World Health Organization (WHO) Global Report-2019, 320,000 people worldwide die every year from drowning as the world's third-leading cause of injury-related death. (Lines: 50-52)

Page 3 line 52 – occur not occurs

Authors responses: Thank you very much for the nice correction. We have corrected accordingly.

Page 4 line 3 – words missing here – suggest two-fifths (42%) of a deaths..

Authors responses: Thank you so much for the nice correction. We have revised accordingly.

Page 4, Line 5 and throughout paper – cross reference the specific panel of the figure when you mention Figure 1 ie Figure 1, Panel A

Authors responses: Thank you very for the suggestion. We have corrected accordingly.

Page 4 line 21 – two-third should be two-thirds – nice should also be nine? I think you mean below 10 years if the age range is 5-9 years inclusive

Authors responses: Thank you very for the points. We have corrected the typos errors. As another category indicates 9 to 14 years age group in reported study (Figure 1, Panel C), the right word is nine, and we followed the original reported report style.⁵

Page 4 line 24 – more victims doesn't make sense, suggest boys are more at risk when compared to girls

Authors responses: Thank you very for the suggestion. We have revised the lines as follows:

The study also reported that boys were significantly more at risk of being victims of drowning when compared to girls (60.82% vs. 38.65%). (Lines: 68-69)

Page 4 – line 39 – are happened to suggest replace with occur

Authors responses: Thank you very for the suggestion. We have corrected accordingly.

Page 4 line 43 – define large households – how many children is considered a large household as per the research?

Authors responses: Thank you very for the questions. We have revised accordingly.

..... children of large households (five or more children) are riskier than children of small families (less than three children)..... (Lines: 79-80)

Page 5 – line 55 – suggest pilot actions not pilot basis actions

Authors responses: Thank you very for the suggestion. We have corrected accordingly.

Page 5 line 5 – focused off – not clear expression – do you mean unfocused or not given enough priority?

Authors responses: Thank you very for the suggestion. We have corrected accordingly.

Page 5 line 10 – a pilot based study or pilot based studies

Authors responses: Thank you very for the suggestion. We have corrected accordingly.

Reviewer: 2

Dr. Colleen Saunders, University of Cape Town Faculty of Health Sciences

Comments to the Author

Dear authors,

Thank you for the opportunity to review this original research letter. I commend you for raising awareness of the drowning burden - particularly in a setting with such a high drowning burden. I have the following comments regarding this submission.

Authors responses: Thank you very much for reviewing the manuscript and providing your opinions for strengthening of the manuscript.

Article format: According to the journal's guidelines for authors, Original Research Letters should present studies that present original research but address a single research question. In my opinion the current submission does not meet this description. The current submission more closely resembles an editorial or letter to the editor that synthesises evidence from several sources and presents these as a narrative review. It does not present original research.

Authors responses: Thank you very much for your observations. We have extracted data from various scholarly databases and recent data available from various media websites. We analyzed and figured the trend and prevalence of the child drownings in Bangladesh that might be considered as an explorative study based on its nature of the data. However, more countrywide epidemiological study needs to be conducted in this regard.

Study title: The title specifically speaks to non-fatal drowning. However the manuscript itself speaks only to the fatal drowning burden in Bangladesh and makes no specific reference to risk factors and interventions related to non-fatal drowning. The title does not, therefore, accurately reflect the contents of the manuscript.

Authors responses: Thank you very much for concerns. We have revised the title and changed it as follows to reflect the study aims.

Child drownings in Bangladesh: Need for action

Language editing: I appreciate that you may not be writing in your first language and commend you for this. However, there are a number of instances in which the grammar and sentence structure is incorrect and changes the meaning of the statement. For example - line 80 - "riskier" versus more at risk; line 67 - "nice" versus nine. I think the manuscript would be strengthened by having a colleague who is fluent in

English provide a brief language editing review.

Authors responses: Thank you very much for your comments and suggestions. We have revised the whole manuscript and eradicated grammar errors through an English language expert. We have also corrected your suggested points.

Specific comments:

- Line 51-52: Please use the updated 2019 WHO estimate (236 000 people) <https://www.who.int/news-room/fact-sheets/detail/drowning>

Authors responses: Thank you very much for your kind suggestions. We have used the updated data according to your suggested link as follows:

According to the World Health Organization (WHO) Global Report-2019, 236000 people worldwide die every year from drowning as the world's third-leading cause of injury-related death.¹

1. World Health Organization. Drowning. Available from: <https://www.who.int/news-room/fact-sheets/detail/drowning> (accessed April 7, 2022)

- Line 65-66: This statement is incorrect - "83% of children died from drowning" should actually read that 83% of drowning victims are children.

Authors responses: Thank you very much for your kind suggestions. We have revised the lines as follows:

Another recent survey performed by SoMaSHTe (Society for Media and Suitable Human Communication Techniques) in support of Global Health Advocacy Incubator (GHAi) documented 1402 deaths from 875 drowning incidents, where 83% (n = 1164) of victims were children in the last one and half years (January 2020 to June 2021).

- Figure 1: Please be cautious of confusing "death rate" with prevalence or "Proportion of all deaths". These are quite different measures and death rate would require a clear indication of the denominator. Authors responses: Thank you very much for your kind suggestions. We have revised and modified the Figure as follows:

Figure 1. (A) The drowning death rate among children aged 1-4 years (2003 and 2011) and below four years (2021) in Bangladesh.^{2,5} (B) The distribution of drowning deaths occurred among Bangladeshi children according to age range in January 2020 to June 2021.⁵ (C) The distribution of child drowning deaths among the divisions of Bangladesh in January 2020 to June 2021.⁵

I wish you well as you take this manuscript forward, and look forward to reading further contributions. Authors responses: Thank you very much for your kind comments.

VERSION 2 – REVIEW

REVIEWER	Reviewer name: Dr. Amy E. Peden Institution and Country: University of New South Wales Sydney, United Kingdom of Great Britain and Northern Ireland Competing interests: None
REVIEW RETURNED	11-Apr-2022

GENERAL COMMENTS	The authors have done a great job in responding to my suggested revisions. I am happy with the revised manuscript.
--

REVIEWER	Reviewer name: Dr. Colleen Saunders Institution and Country: University of Cape Town Faculty of Health Sciences, South Africa
REVIEW RETURNED	19-Apr-2022

GENERAL COMMENTS	Thank you for re-submitting this manuscript for review after engaging with the reviewers comments and suggestions. I think that you have for the most part addressed the previous comments. The only remaining major comment is that there still remains some confusion between period prevalence or proportion of deaths (indicated with a %) versus death rates (which require a denominator value e.g. x drowning deaths per 100 000 population), in Figure 1. As it currently stands, Figure 1 A is not easy to interpret. I also have the following specific comments:  - Figure 1 A & Page 2 Line 54-56: Please differentiate between drowning death prevalence (%) and drowning death rates (which are presented as a fraction i.e. x per y). Figure 1 presents a period frequency and prevalence, not drowning rate. - Page 2, line 55: Suggest correct as follows ".two-fifths (42%) of all deaths in children aged 1 to 4 years..." - Page 3, line 77: I wouldn't consider "buckets" as natural water bodies. - Page 4 Line 94: "children" not "kids"
--

VERSION 2 – AUTHOR RESPONSE

Date: 29-Apr-2022

Manuscript ID: bmjpo-2022-001464.R1

Manuscript Title: Child drownings in Bangladesh: Need for action

Editor in Chief Comments to Author :

Replace legend for Figure 1A "The drowning death rate among children aged 1-4 years (2003 and 2011) and below four years (2021) in Bangladesh." with "Percentage of drowning deaths in children aged 1-4 years (2003 and 2011) and in children under 4 years of age(2021) in Bangladesh"

Authors' responses: Dear Respected Editor-in-Chief, we would like to thank you and your assigned reviewers for reviewing the manuscript in second round and providing your expert opinions & suggestions for the improvement of the manuscript. The figure 1A legend has been changed as follows:

Figure 1. (A) Percentage of drowning deaths in children aged 1-4 years (2003 and 2011) and in children under 4 years of age (2021) in Bangladesh.^{2,5}(B) The distribution of drowning deaths occurred among Bangladeshi children according to age range in January 2020 to June 2021.⁵ (C) The distribution of child drowning deaths among the divisions of Bangladesh in January 2020 to June 2021.⁵

Line 58 Replace "Besides, the prevalence of death rate from the drowning of rural children is significantly higher than urban children in the country" with "The risk of drowning is significantly higher in rural children"

Authors' responses: Thank you very much for your kind suggestions. We have replaced accordingly as follows:

The risk of drowning is significantly higher in rural children. (Line: 58)

Line 70 Replace "prevalence in terms of death rates in the country" with "number of deaths"

Authors' responses: Thank you very much for the suggestion. We have replaced it accordingly.

Line 77 delete "natural water bodies like "

Authors' responses: Thank you very much for the suggestion. We have deleted it accordingly.

Line 83 replace "parenteral" with "parental"

Authors' responses: Thank you very much for the correction. We have replaced it accordingly.

Reviewer: 1

Dr. Amy E. Peden, University of New South Wales Sydney

Comments to the Author

The authors have done a great job in responding to my suggested revisions. I am happy with the revised manuscript.

Authors' responses: I would like to thank you very much for reviewing our manuscript and providing your kind feedback.

Reviewer: 2

Dr. Colleen Saunders, University of Cape Town Faculty of Health Sciences

Comments to the Author

Thank you for re-submitting this manuscript for review after engaging with the reviewers comments and suggestions. I think that you have for the most part addressed the previous comments.

The only remaining major comment is that there still remains some confusion between period prevalence or proportion of deaths (indicated with a %) versus death rates (which require a denominator value e.g. x drowning deaths per 100 000 population), in Figure 1. As it currently stands, Figure 1 A is not easy to interpret.

I also have the following specific comments:

- Figure 1 A & Page 2 Line 54-56: Please differentiate between drowning death prevalence (%) and drowning death rates (which are presented as a fraction i.e. x per y). Figure 1 presents a period frequency and prevalence, not drowning rate.

Authors' responses to both comments: I would like to thank you very much for reviewing our manuscript in the second round and providing your kind opinions and suggestions for the improvement of the manuscript. We have changed the legend of the Figure 1A and changed the Y-axis title of the Figure to eradicate the confusion according to the suggestion of the Editor-in-Chief as follows:

Figure 1. (A) Percentage of drowning deaths in children aged 1-4 years (2003 and 2011) and in children under 4 years of age (2021) in Bangladesh.^{2,5}(B) The distribution of drowning deaths occurred among Bangladeshi children according to age range in January 2020 to June 2021.⁵ (C) The distribution of child drowning deaths among the divisions of Bangladesh in January 2020 to June 2021.⁵

Also, we have changed in the main text.

- Page 2, line 55: Suggest correct as follows "..two-fifths (42%) of all deaths in children aged 1 to 4 years..."

Authors responses: Thank you very much for the suggestion. We have replaced it accordingly.

- Page 3, line 77: I wouldn't consider "buckets" as natural water bodies.

Authors responses: Thank you very much for the suggestion. We have revised the line as follows:

Around 80% of the drowning deaths occur due to exposure to ponds, channels, buckets, and ditches within 20 meters of victim's home. (Lines: 75-77)

- Page 4 Line 94: "children" not "kids"

Authors' responses: Thank you very much for the suggestion. We have replaced it accordingly.

VERSION 3 – REVIEW

REVIEWER	Reviewer name: Institution and Country: Competing interests:
REVIEW RETURNED	

GENERAL COMMENTS	
--

REVIEWER	Reviewer name: Institution and Country: Competing interests:
REVIEW RETURNED	

GENERAL COMMENTS	
--

REVIEWER	Reviewer name: Institution and Country: Competing interests:
REVIEW RETURNED	

GENERAL COMMENTS	
--

VERSION 3 – AUTHOR RESPONSE